# Effects of Different Types of Chronic Training on Bioenergetic Profile and Reactive Oxygen Species Production in LHCN-M2 Human Myoblast Cells

**DOI:** 10.3390/ijms23147491

**Published:** 2022-07-06

**Authors:** Annamaria Mancini, Daniela Vitucci, Giuseppe Labruna, Stefania Orrù, Pasqualina Buono

**Affiliations:** 1Department of Movement Sciences and Wellness, University Parthenope, 80133 Napoli, IT, Italy; annamaria.mancini@uniparthenope.it (A.M.); vitucci@ceinge.unina.it (D.V.); stefania.orru@uniparthenope.it (S.O.); 2CEINGE-Biotecnologie Avanzate, 80145 Napoli, IT, Italy; 3IRCCS SDN, 80143 Napoli, IT, Italy; giulabruna@hotmail.it

**Keywords:** oxidative stress, metabolism, exercise, human myoblast LHCN-M2, longevity

## Abstract

Human skeletal muscle contains three different types of fibers, each with a different metabolism. Exercise differently contributes to differentiation and metabolism in human myoblast cells. The aims of the present study were to investigate the effects of different types of chronic training on the human LHCN-M2 myoblast cell bioenergetic profile during differentiation in *real time* and on the ROS overproduction consequent to H_2_O_2_ injury. We demonstrated that exercise differently affects the myoblast bioenergetics: aerobic exercise induced the most efficient glycolytic and oxidative capacity and proton leak reduction compared to untrained or anaerobic trained sera-treated cells. Similarly, ROS overproduction after H_2_O_2_ stress was lower in cells treated with differently trained sera compared to untrained sera, indicating a cytoprotective effect of training on the reduction of oxidative stress, and thus the promotion of longevity. In conclusion, for the first time, this study has provided knowledge regarding the modifications induced by different types of chronic training on human myoblast cell bioenergetics during the differentiation process in *real time*, and on ROS overproduction due to stress, with positive implications in terms of longevity.

## 1. Introduction

Exercise is considered a non-pharmacological tool that contributes to the quality of mitochondrial machinery and the efficiency of skeletal muscle, improving overall muscle health [1]. Human skeletal muscle contains three different types of fibers: type I, slow-twitch fiber, with oxidative metabolism; type II a and b fibers, fast-twitch fibers, with a predominant glycolytic metabolism; and intermediate fibers with glycolytic/oxidative metabolism [2,3]. Mitochondria are highly active organelles, the main site of oxidative metabolism, and consequently of reactive oxygen species (ROS) production [4,5]. Moderate, regular exercise training affects the number and size of mitochondria in skeletal muscle and the metabolic profile of fibers, with an enrichment of oxidative fibers [6,7,8,9]; chronic exercise training adaptation, conversely, results in a reduction of ROS production, also during an acute bout [10,11,12]. A sedentary lifestyle or ageing are associated with impaired respiration and elevated ROS production in skeletal muscle fibers [1,13,14,15].

Although small concentrations of ROS are required in cellular signaling, their overproduction is the major determinant of oxidative stress [16]. Many pathological processes, such as neurodegenerative diseases or ageing, are associated with a derangement between ROS generation and antioxidant capacity [17,18]. Furthermore, high levels of ROS, together with an increase of reactive nitrogen species (RNS), activate an apoptosis pathway associated with ageing [19,20,21].

Tissue metabolism can be investigated in *real time* through the measure of the rate of oxygen consumed in cultured cells using the oxygen consumption rate (OCR) and the extracellular acidification rate (ECAR) indicators, as reported in Nicholls et al. [22]. Until now, Rat L6 and, above all, mouse C2C12 myoblast cells, have been used to evaluate the effects of contraction/exercise-like stimuli on muscle cell metabolism [23,24,25].

The human myoblast LHCN-M2 cell line represents the first species-specific system of human skeletal muscle immortalized cells able to differentiate into mature myotubes. We recently demonstrated that different types of sport training promote LHCN-M2 myoblast differentiation, inducing the expression of early and late myogenic markers, together with molecular markers associated with oxidative metabolism [26]. No data have been provided so far on the effects mediated by different types of chronic training on human muscle cell bioenergetics during myoblast differentiation in *real time*. Thus, the aims of the present study were to investigate the effects of different types of chronic training on the human LHCN-M2 myoblast cell bioenergetic profile during differentiation using the Seahorse XFe96 analyzer [27], and to test the protective effect mediated by different types of training against ROS overproduction consequent to H_2_O_2_ injury in this system.

## 2. Results

### 2.1. Real Time Determination of the Effects Mediated by Different-Types of Training on the Bioenergetic Profile in Human LHCN-M2 Myoblast Cells during Differentiation

The clinical–biochemical characteristics of enrolled subjects are reported in Table 1. To elucidate the effect of different types of training on the metabolic profile of human myoblasts during differentiation, we analysed, in *real time*, the glycolytic pathway component by ECAR (Figure 1A), and the oxidative phosphorylation rate, measured by OCR (Figure 2A). Both bioenergetic profiles resulted higher in proliferating LHCN-M2 cells (GM, growth medium) compared to cells induced to differentiation using human pool sera from differently trained volunteers or differentiation medium (DM). In particular, the glycolytic capacity was lower in cells treated with sera from anaerobic (body builders, BB) compared to aerobic (swimmers, SW; *p* <0.05) trained volunteers, or untrained (UN; *p* < 0.01) or DM (*p* < 0.01); moreover, glycolytic capacity was also lower in cells treated with sera from aerobic/anaerobic (soccer players, SO + volleyball players, VB) trained volunteers compared to UN (*p* < 0.01) or DM (*p* < 0.01) (Figure 1B).

Basal (Figure 2B) and maximal (Figure 2E) respiration, ATP-linked respiration (Figure 2C), proton leak (Figure 2D) and spare respiratory capacity (Figure 2F) were all higher in proliferative cells (GM) compared to cells treated with pool sera from differently trained volunteers (SO + VB; SW; BB), or UN or in DM treated cells (*p* < 0.001).

The ATP-linked respiration correlates inversely to the mitochondrial proton leak; in fact, it was higher in cells treated with pool sera from “mixed” sports (SO + VB) compared to anaerobic (BB) (*p* < 0.01) or UN (*p* < 0.001). The ATP-linked respiration was also higher in aerobic (SW) volunteers compared to UN (*p* < 0.01), and in anaerobic (BB) volunteers compared to UN (*p* < 0.05) (Figure 2C). Conversely, proton leak is lower in cells treated with sera from differently trained volunteers compared to UN (SO + VB, *p* < 0.001; SW, *p* < 0.01; and BB, *p* < 0.05 vs. UN; Figure 2D).

### 2.2. Effects of Sera from Differently Trained Volunteers on LHCN-M2 Cells and ROS Production

The 5 μmol/L H_2_O_2_ treatment affected the ROS generation differently in LHCN-M2 cells treated with pool sera from differently trained volunteers compared to UN. In particular, the production of ROS was lower in cells treated with sera from mixed (SO + VB; *p* < 0.001) or anaerobic (BB; *p* < 0.01) or aerobic (SW; *p* < 0.01) trained volunteers compared to UN. No differences were found in cells cultured in DM compared to UN (Figure 3A).

The production of ROS was further reduced in cells treated with 10 μmol/L H_2_O_2_ and cultured with SO+VB (*p* < 0.01), BB (*p* < 0.001) and SW (*p* < 0.01) sera compared to DM or UN sera (Figure 3B), showing a dose-effect.

## 3. Discussion

In mammalian cells, oxidative phosphorylation and glycolytic pathways provide for the synthesis of cellular ATP. The ATP synthesis/degradation is regulated by the ATP/ADP ratio in muscle cells. The metabolism of precursor muscle cells is differently regulated during myoblast differentiation. Quiescent muscle satellite cells (SCs) show a low-rate metabolism and a TCA cycle responsible mostly for ATP production in the quiescent state [28,29]. After the activation, SCs shift from lipid, oxidative, to glycolytic metabolism [30]. The shift toward glycolytic pathways in C2C12 myoblasts during differentiation was associated with pyruvate production and mTOR and P70S6K phosphorylation [31].

Exercise and muscle contraction activate SCs proliferation and differentiation in mature myotubes; during this shift, the energetic demand is supported by aerobic/anaerobic metabolism [32].

Previously, we demonstrated that chronic aerobic exercise, such as swimming, induces highest fusion index percentage and muscle-specific late differentiation markers, i.e., MyHC-β expression, in LHCN-M2 human myoblast compared to untrained, anaerobic, or mixed sport treated cells. Conversely, cells treated with sera from anaerobic trained volunteers, such as body builders, showed a higher expression of early differentiation markers, including CK activity and myogenin, and a lower fusion index percentage. The expression of MyHC-β in LHCN-M2 cells treated with sera from aerobic exercised volunteers were associated with the increased expression of molecular markers of oxidative metabolism compared to cells cultured in DM or treated with untrained sera [26].

To the best of our knowledge, this is the first study to use human myoblast cell lines to test the effects of different types of chronic training on the myoblast bioenergetics in *real time*.

As expected, the measure of both ECAR and OCR profiles resulted in higher proliferation compared to differentiated LHCN-M2 cells, due to the higher metabolic rate in proliferating cells. Interestingly, after oligomycin injection, leading to the inhibition of ATP synthase (Complex V), the glycolytic capacity, measured as acidification of medium (ECAR), was lower in cells treated with sera from differently trained volunteers compared to untrained or DM. Our results suggest a metabolic adaptation induced by long-term training in terms of the most efficient use of the energetic membrane potential, highest ATP-linked respiration and lowest proton leak, in cells treated with sera from aerobic or mixed sport trained volunteers compared to cells treated with sera from untrained or anaerobic sport trained volunteers.

These observations were further supported by the increased expression of cellular biomarkers associated with the oxidative metabolism found in cells treated with aerobic or mixed exercised volunteers compared to anaerobic volunteers or untrained sera [26].

Previous studies investigated the effects mediated by acute or chronic exercise on respiratory and oxidative capacity, indicating a reduction in ROS production mediated by chronic exercise [10,11,33]. On the contrary, muscle diseases induce an increase in ROS production and an activation of the apoptosis pathways, leading to muscle atrophy [34,35,36,37].

Furthermore, detoxification mechanisms balance ROS levels in healthy cells [38], and mitochondrial ROS generation is a significant part of the ageing process [39,40], increasing with age in mammalian mitochondria [41,42]. ROS are generally considered the primary cause of macromolecular damage [43]; however, when produced in a controlled manner, ROS play important roles in cell signaling [44]. There is a close relationship between proton leak and ROS generation [45,46]. Proton leak and ROS overproduction also represent the substrate for different cardiovascular, neurodegenerative and metabolic (obesity, diabetes) diseases and for the ageing process [47]. 

Here, we have demonstrated that LHCN-M2 human myoblast cells treated with sera from differently trained volunteers reduced ROS production in a dose-dependent manner after H_2_O_2_ addition, in line with previous studies indicating the cytoprotective effects of long-term training on the reduction of oxidative stress and on longevity promotion [48,49].

All things considered, our results suggest the presence of specific biomarkers in the sera of differently long-term trained volunteers are able to influence human myoblast cell bioenergetics and counteract oxidative stress, promoting longevity. From this perspective, we pursued the identification of the biomarkers circulating in the serum of differently trained volunteers that could be involved in longevity pathways.

The LHCN-M2 human immortalized myoblast cells represent a very promising homologous system, mirroring the skeletal muscle. These cells represent a useful tool for studying myogenic differentiation, and also the molecular effects mediated by different types of exercise, on metabolism, bioenergetic profile, oxidative stress protection and on sarcopenia determinants. In terms of clinical implications, this system could be used to develop therapeutic applications in sport-associated skeletal muscle lesions or in the identification of molecular markers involved in dysmetabolic diseases and longevity promotion for the development of novel diagnostic therapies. The clinical limitations of this system are the same as cellular system use; in the future, it may be interesting to perform these experiments in primary myoblast cells or in animal models.

## 4. Materials and Methods

### 4.1. Cell Culture and Treatments

LHCN-M2, human skeletal myoblasts immortalized cells [50], derived from pectoralis major muscle of a 41-year-old Caucasian man, were obtained by Dr. Vincent Mouly (Institut de Myologie, Paris, France). Proliferating LHCN-M2 cells were maintained in growth medium (GM) containing 15% heat inactivated fetal bovine serum (FBS) at 37 °C in 5% CO_2_ at sub-confluent density (70%), and induced to differentiate in differentiation medium (DM) supplemented with 0.5% of heat-inactivated FBS for 4 d. GM and DM are detailed in Vitucci et al. [26].

### 4.2. Subject Recruitment

Blood sample collection of healthy participants was detailed in Vitucci et al. [26]. 

Young healthy adult (18–28 y) male volunteers—N = 6 swimmers (SW), N = 6 soccer players (SO), N = 6 Body builders (BB) and N = 6 volleyball players (VB)—who had played sports at least in the last 3 years for ≥180 min/week, were recruited. Blood samples were collected in the morning, at least 8 h after the last training session. Clinical–biochemical characterization of the enrolled volunteers is reported in Table 1. Sera samples belonging from volunteers practicing the same sport were pooled and immediately stored at −80 °C until cell treatment.

All subjects gave their informed consent for inclusion before they participated in the study. The study was conducted in accordance with the Declaration of Helsinki, and the protocol was approved first by the Ethics Committee of the SDN-IRCSS, Naples, and successively by the Ethics Committee of the University of Naples “Federico II” (protocol code n. 207/19).

### 4.3. LHCN-M2 Treatment and Bioenergetics Profiling

Glycolysis and mitochondrial respiration were analyzed in LHCN-M2 cells treated with sera pool from differently trained volunteers during differentiation in *real time* by measuring the extracellular acidification rate (ECAR) and the oxygen consumption rate (OCR), as detailed in Omodei et al. 2015 and Iaffaldano et al. 2018 [51,52], using the Seahorse XFe96 (Seahorse Bioscience, North Billerica, MA) instrument.

For each condition tested, LHCN-M2 cells were maintained in GM on 96-well Seahorse plates at 8 × 10^3^ cells/well. Differentiation was induced in DM supplemented with 0.5% heat inactivated pool sera from SW, BB, SO+VB or 0.5% lyophilized sera pool from untrained healthy subjects (control, UN; Randox Laboratories Ltd., Crumlin, UK), for 4 days. Cell growth in GM or DM represent experimental controls. ECAR was measured in XF media in basal conditions and after 10 mM glucose, 1 mM oligomycin and 50 mM 2-deoxy-D-glucose (Glycolysis Inhibitor, 2-DG) addition by using the XF glycolysis stress test kit (Agilent Technologies, Craven Arms, UK), according to the manufacturer’s instructions.

OCR was measured in basal conditions and in response to 1 mM oligomycin (ATP synthase inhibitor, Olig), 1 mM of carbonylcyanide-4-(trifluoromethoxy)-phenylhydrazone (mitochondrial decoupling, FCCP) and 1 mM of antimycin A (inhibitor of Complex III, Ant) and rotenone (Complex I inhibitor, Rot) by using the XF Mito Stress Test Kit (Agilent Technologies, Craven Arms, UK), according to the manufacturer’s instructions. The measure of ECAR and OCR were performed three times, approximately every 10 min.

### 4.4. Measurement of Intracellular ROS

Intracellular ROS production was obtained using 2′,7′-dichlorofluorescin diacetate (DCFDA)-cellular ROS Assay Kit (abcam, Cambridge, UK) in accordance to the manufacturer’s instructions. LHCN-M2 cells were grown in 96-well plates at 3 × 10^3^ cells/well in DM, with 0.5% pool sera from SW, BB, SO + VB or UN, for 4 days; cells cultured with GM and DM represent the experimental control. After 4 days of treatment, growth media were replaced with GM containing 5 or 10 μmol/L of fresh hydrogen peroxide, respectively. After 2 h, the cells were washed with 1X buffer (provided in the kit), then incubated with DFCDA 10 μM for 45 min at 37 °C in the dark, and washed again with 1× buffer. Dichlorofluorescein (DCF) production was measured by a PerkinElmer VICTOR3 at an Ex-485 nm and Em-535 nm.

### 4.5. Statistical Analysis

Data were expressed as mean ± SD or SEM as appropriate. Data comparison between groups was performed using the Kruskal–Wallis test or ANOVA test, as appropriate. Differences were considered statistically significant with a *p* < 0.05 after Bonferroni or Fisher correction, as appropriate. Statistical analyses were carried out with the PASW package for Windows (ver.18; SPSS Inc. Headquarters, Chicago, IL) and *StatView* software (version 5.0.1.0; SAS Institute Inc., Cary, NC, USA).

## 5. Conclusions

In conclusion, for the first time, this study has provided knowledge regarding the modifications induced by different types of long-term training on human myoblast cell bioenergetics during the differentiation process in *real time*, and on ROS overproduction due to stress with implications in terms of longevity.

## Figures and Tables

**Figure 1 ijms-23-07491-f001:**
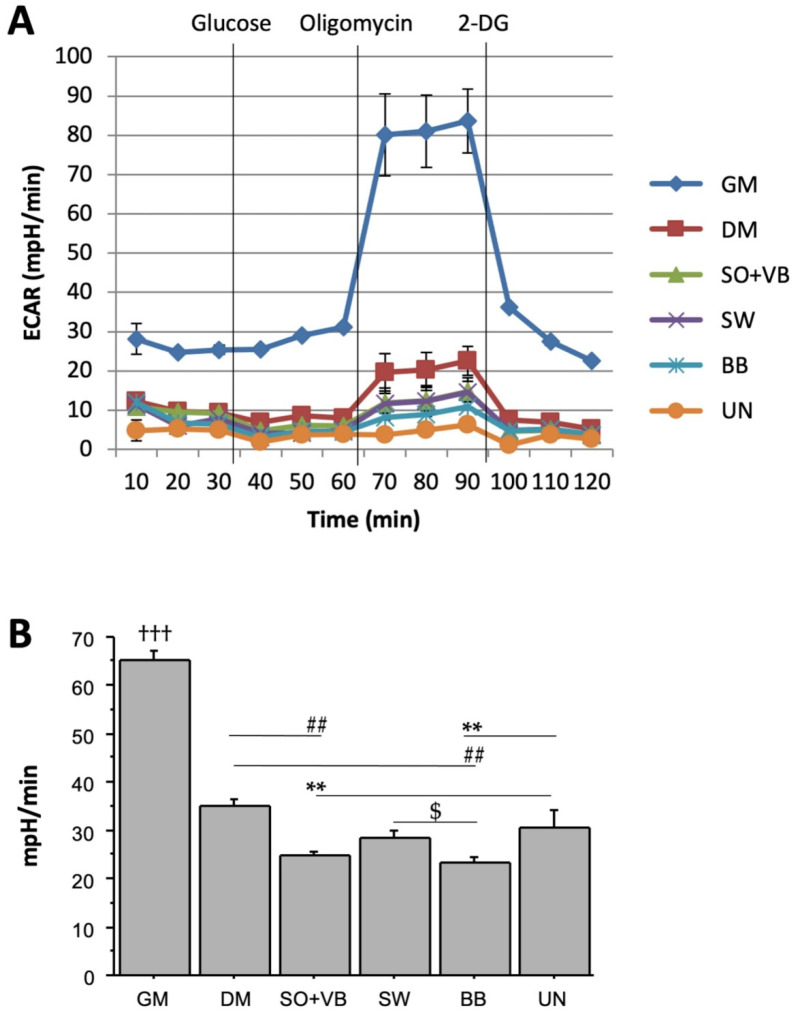
Extracellular acidification rate (ECAR) measured in *real time* under basal condition and in response to glucose, oligomycin and 2-deoxy-D-glucose (2DG) in differently trained sera-treated LHCN-M2 cells. (**A**) Bioenergetic profiling of differently treated LHCN-M2 cells. (**B**) Glycolytic capacity measured after the addition of oligomycin. Data expressed as mean ± SEM of four independent experiments. Data comparison between groups were performed using the Student’s *t*-test. ††† *p* < 0.001 GM vs. all treated cells; ** *p* < 0.01 vs. UN; ## *p* < 0.01 vs. DM; $ *p* < 0.01 SW vs. BB.

**Figure 2 ijms-23-07491-f002:**
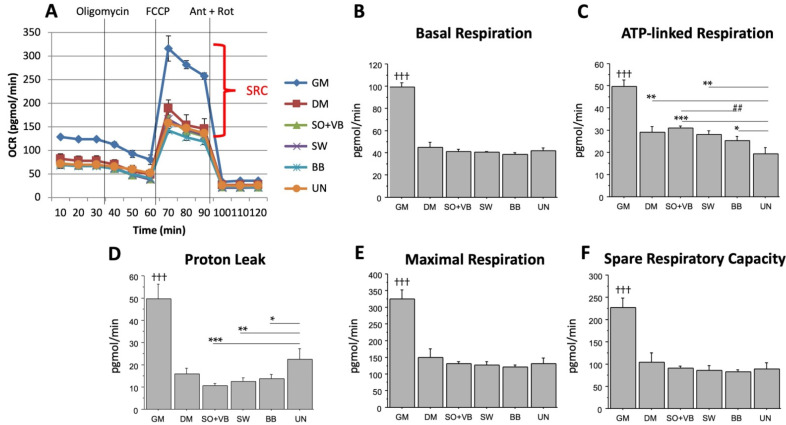
Oxygen consumption rate (OCR) measured in differently trained sera-treated LHCN-M2 cells in *real time* under basal conditions and in response to the following inhibitors: oligomycin, carbonylcyanide-4-(trifluoromethoxy)-phenylhydrazone (FCCP), antimycin A and rotenone (Ant + Rot). (**A**) Bioenergetic profile; SRC (spare respiratory capacity) in GM cultured cells is reported for explanatory purposes; (**B**) basal respiration (before addition of oligomycin); (**C**) ATP-linked respiration (calculated as the difference between basal and after oligomycin addition rate); (**D**) proton leak (calculated as difference between oligomycin and Ant + Rot rate); (**E**) maximal respiration (calculated as the difference between FCCP and Ant + Rot rate); and (**F**) spare respiratory capacity (calculated as the difference between maximal respiration and basal respiration). Data expressed as mean ± SEM of four independent experiments. Data comparison between groups was performed using the Student’s *t*-test. ††† *p* < 0.001 GM vs. all treated cells; *** *p* < 0.001; ** *p* < 0.01; * *p* < 0.05 vs. UN; ## *p* < 0.01 vs. SO + VB.

**Figure 3 ijms-23-07491-f003:**
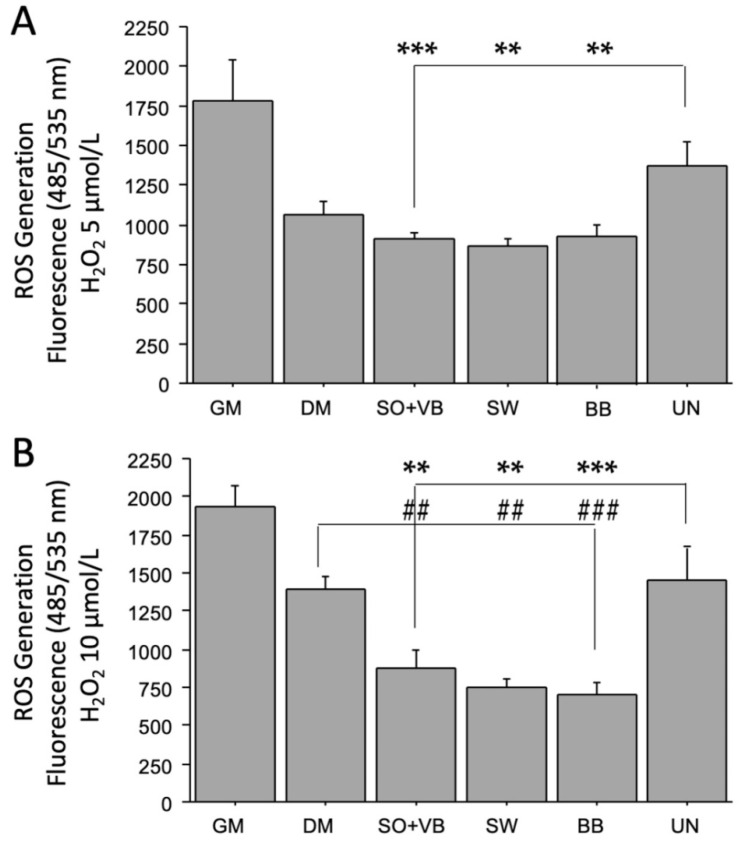
Reactive oxygen species (ROS) measured by the 2′,7′-dichlorofluorescin diacetate (DCFDA) assay. Sera-treated LHCN-M2 cells were insulted with increased H_2_O_2_ concentrations: (**A**) 5 μmol/L, (**B**) 10 μmol/L. Data comparison between groups was performed using the Student’s *t*-test. Data expressed as mean ± SEM of three independent experiments. *** *p* < 0.001 or ** *p* < 0.01 vs. UN; ### *p* < 0.001 or ## *p* < 0.01 vs. DM.

**Table 1 ijms-23-07491-t001:** Clinical–biochemical characterization of enrolled volunteers.

Sport	Age (y)	Height (m)	Weight (kg)	BMI (kg/m^2^)	IGF-1 * ng/mL	Glucose mg/dL	LDH U/L	Hemoglobin (Hb) g/dL
				18.5–24.9 kg/m^2^	116–358 ng/mL	60–110 mg/dL	227–450 U/L	13–17.5 g/dL
SO	22.8 ± 1.9	1.8 ± 0.1	71.2 ± 6.4	23.0 ± 1.6	230.7 ± 117.6	87.8 ± 11.6	380.4 ± 36.5	16.2 ± 0.9
VB	24.3 ± 1.0	1.8 ± 0.0	72.3 ± 5.0	23.0 ± 1.3	241.0 ± 92.9	84.0 ± 18.7	372.5 ± 38.1	15.9 ± 0.6
SW	22.7 ± 3.9	1.7 ± 0.0	69.2 ± 4.9	22.9 ± 1.3	190.2 ± 50.2	80.2 ± 10.5	400.3 ± 21.9	13.9 ± 1.7
BB	25.7 ± 2.4	1.8 ± 0	73.3 ± 7.2	23.5 ± 1.3	227.0 ± 39.3	79.5 ± 14.9	431.5 ± 21.7	15.2 ± 0.8
	Total Cholesterol mg/dL	HDL Cholesterol mg/dL	Triglycerides mg/dL	AST U/L	ALT U/L	GGT U/L	Sideremia µg/dL	Hematocrit (Ht) %
	100–200 mg/dL	>35 mg/dL	<150 mg/dL	1–40 U/L	1–40 U/L	1–50 U/L	53–167 μg/dL	42.0–50.0%
SO	177.3 ± 37.5	48.4 ± 9.6	127.8 ± 57.1	29.6 ± 10.2	32.0 ± 1.0	35.8 ± 17.5	125.6 ± 31.0	48.9 ± 2.3
VB	143.0 ± 24.5	49.0 ± 3.7	128.0 ± 81.6	21.0 ± 2.4	28.7 ± 11.2	33.3 ± 15.4	90.0 ± 22.0	48.8 ± 1.6
SW	166.2 ± 4.1	55.5 ± 12.3	82.5 ± 46.3	22.7 ± 5.9	23.2 ± 11.6	30.2 ± 19.4	89.5 ± 44.5	43.0 ± 4.1
BB	184.7 ± 25.5	50.7 ± 17	87.7 ± 20.8	30.2 ± 6.6	31.2 ± 4.0	13.8 ± 2.9	126.7 ± 58.2	45.2 ± 2.9

* IGF-1 = 88 ng/mL in UN pool of human sera from untrained healthy subjects. Abbreviations: BMI: body mass index; IGF-1: insulin-like growth factor-1; LDH: lactate dehydrogenase; AST: aspartate aminotransferase; ALT: alanine aminotransferase; GGT: gamma-glutamyltransferase; SO: soccer players; VB: volleyball players; SW: swimmers; BB: body builders. Data are expressed as mean ± SEM. Table modified from Vitucci et al. [26].

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
