# Peer review of "Effects of Different Types of Chronic Training on Bioenergetic Profile and Reactive Oxygen Species Production in LHCN-M2 Human Myoblast Cells"

_ijms, 2022, doi:10.3390/ijms23147491_

Round 1

Reviewer 1 Report

A paper titled Effects of different type of chronic training on bioenergetic profile and Reactive Oxygen Species production in LHCN-M2 human myoblast cells. Whose aim was of the present study were to investigate the effects of chronic different types of training on the human LHCN-M2 myoblast cell bioenergetic profile during differentiation in real time and on the ROS overproduction consequent to H2O2 injury.

It is well written. I notice only minor editing errors. Congrats to the authors on the paper and the idea.

 Please standardize the font (abstract, table, main text). The bibliography is not in MPDI style please correct it. 

Author Response

Dr. José L. Quiles

Department of Physiology, Institute of Nutrition and Food Technology “Jose Mataix”, Biomedical Research Center, University of Granada, Avda. Conocimiento s/n, 18100 Armilla, Granada, Spain

Section Editor-in-Chief: International Journal of Molecular Sciences; Section: “Molecular Endocrinology and Metabolism”

Ms. No. ijms-1776633
Effects of chronic different type of training on bioenergetic profile and Reactive Oxygen Species production in LHCN-M2 human myoblast cells

Dear Dr Josè Quiles,

Thank you for your decision letter regarding the above-referenced original research article that we submitted to the International Journal of Molecular Sciences. We were very pleased to learn that you are prepared to re-consider our article provided we address all the concerns raised by the reviewers. We found the reviewers’ comments and suggestions very helpful and have revised the article accordingly.

Further, according to Dr. Qian Yuan, assistant editor suggestions, we revised the section 4, Materials and Methods.

We are therefore resubmitting the tracked revised paper, together with an item-by-item response to each of the comments made. 

Looking forward to hearing from you,

Yours sincerely,

Pasqualina Buono, Ph.D.

Dipartimento di Scienze Motorie e del Benessere

Università di Napoli Parthenope

Via Medina 40

80133 Napoli, Italy

Tel.: +39 081 5474674

e-mail: buono@uniparthenope.it

Ms. No. ijms-1776633. Title: Effects of chronic different type of training on bioenergetic profile and Reactive Oxygen Species production in LHCN-M2 human myoblast cells

Reviewer #1:

A paper titled Effects of different type of chronic training on bioenergetic profile and Reactive Oxygen Species production in LHCN-M2 human myoblast cells. Whose aim was of the present study were to investigate the effects of chronic different types of training on the human LHCN-M2 myoblast cell bioenergetic profile during differentiation in real time and on the ROS overproduction consequent to H2O2 injury.

It is well written. I notice only minor editing errors. Congrats to the authors on the paper and the idea.

Please standardize the font (abstract, table, main text). The bibliography is not in MPDI style please correct it. 

We thank the reviewer#1 for the observation. We standardize the font and modified the bibliography  accordingly the MDPI style.

Reviewer #2:

This study reported knowledge on the modifications induced by different types of chronic training on the human myoblast cells bioenergetic during the differentiation process in real-time and on the ROS generation. This study is interesting and well signed. However, there are several issues that need to be addressed. A revision is suggested.

  1. It is unclear why this cell line was used.

We thank the reviewer#2 for the observation that give us the possibility to further clarify this point.

The LHCN-M2 are, to our best knowledge, the only immortalized human myoblast cells capable of overcoming cellular senescence through the expression of the telomerase reverse transcriptase and cyclin-dependent kinase 4 (Zhu et al., 2007). We previously set-up the best cellular growth and differentiation conditions for myoblast differentiation process in these cells. Recently, we used these cells in order to study the effects induced by sera from differently-trained volunteers on Fusion Index and on the expression of early and late markers of muscle-specific differentiation, i.e. myogenin and MyHC-β (Vitucci et al. 2019). Here, we aimed to unravel the effects mediated by different-type of training on human myoblast cells bioenergetic during differentiation by using the same in-vitro homologous system, human LHCN-M2 cells and human sera.

We evidenced the use of these cells in the paper: Section 1.: Introduction, pag 2, line 57: Human myoblast LHCN-M2 cell line represents the first species-specific system of human skeletal muscle immortalized cells able to differentiate into mature myotubes. We recently demonstrated that different type of sport training promote LHCN-M2 myoblast differentiation inducing early and late myogenic markers expression together with molecular markers associated to oxidative metabolism [26].   

In Discussion section; page 8, line 163: To the best of our knowledge this is the first study to use human myoblast cell line to test the effects of different type of chronic training on the myoblast bioenergetics in real time.

  1. Please provide the information on IRB approval.

We thank the reviewer#2 for the observation that give us the possibility to further clarify this point.

As suggested by the reviewer, we have added the information on IRB approval in the "Subject recruitment" paragraph, page 9, according to the instructions of the authors. We provide further information about the IRB, now read: All subjects gave their informed consent for inclusion before they participated in the study. The study was conducted in accordance with the Declaration of Helsinki, and the protocol was approved first by Ethics Committee of the SDN-IRCSS, Naples and successively by the Ethics Committee of the University of Naples “Federico II” (protocol code n. 207/19)”.

  1. It is unclear how to calculate the subject number.

We thank the reviewer#2 for the observation that give us the possibility to further clarify this point.

In this study we used the same pool sera as in our previous study (Vitucci et al .2019). Here, we focused on the effects induced by sera from differently-trained volunteers on the human LHCN-M2 myoblast cells bioenergetics during differentiation. Hence, in order to minimize the biological differences that could affect the cellular biological response and to maximize the effects determined by different sport, we pooled the sera from volunteers belonging to the same sports categories having similar biochemical-clinical characteristics and volume of physical activity.

  1. Please discuss the limitations and strengths of the clinical implications.

We thank the reviewer#2 for the observation that give us the possibility to complete this point

According to reviewer#2 suggestions, we added in the Discussion section, pag 9 the following paragraph: The LHCN-M2 human immortalized myoblast cells represent a very promising homologous system mirroring the skeletal muscle. These cells represent an useful tool in order to study myogenic differentiation and also the molecular effects mediated by different types of exercise, on metabolism, bioenergetic profile, oxidative stress protection and on sarcopenia determinants. In terms of clinical implications, this system could be used to develop therapeutic applications in sport-associated skeletal muscle lesions or in the identification of molecular markers involved in dysmetabolic diseases and longevity promotion for the development of novel diagnostic therapies. The clinical limitations of this system are the same of cellular system use; in the future may be interesting to perform these experiments in primary myoblast cells or in animal model.

5.Please consult a writing specialist to assist with grammar and writing structure.

A professional statistician is also highly recommended to conduct your data analyses.

As suggested by the reviewer #2, we entrusted the grammar and writing structure review by a native English speaker, we acknowledged in the paper.

Statistical evaluations were conducted by two independent researchers, with two different statistical tools (prisma and SPSS), as reported in Materials and Methods section, with the support of the Bioinformatics Unit of CEINGE, Biotecnologie avanzate, Napoli, Italy.

Reviewer 2 Report

This study reported knowledge on the modifications induced by different types of chronic training on the human myoblast cells bioenergetic during the differentiation process in real-time and on the ROS generation. This study is interesting and well signed. However, there are several issues that need to be addressed. A revision is suggested.

1.    It is unclear why this cell line was used.

2.    Please provide the information on IRB approval.

3.    It is unclear how to calculate the subject number.

4.    Please discuss the limitations and strengths of the clinical implications.

5.    Please consult a writing specialist to assist with grammar and writing structure. A professional statistician is also highly recommended to conduct your data analyses.

Author Response

(The authors gave the same response as above.)

Round 2

Reviewer 2 Report

My questions had been well addressed, this submission is acceptable.